# Longitudinal Characterization of the Mumps-Specific HLA-A2 Restricted T-Cell Response after Mumps Virus Infection

**DOI:** 10.3390/vaccines9121431

**Published:** 2021-12-03

**Authors:** Josien Lanfermeijer, Marieke M. Nühn, Maarten E. Emmelot, Martien C. M. Poelen, Cécile A. C. M. van Els, José A. M. Borghans, Debbie van Baarle, Patricia Kaaijk, Jelle de Wit

**Affiliations:** 1Center for Infectious Disease Control, National Institute for Public Health and the Environment, 3721 MA Bilthoven, The Netherlands; josien.lanfermeijer@rivm.nl (J.L.); M.M.Nuhn-3@umcutrecht.nl (M.M.N.); maarten.emmelot@rivm.nl (M.E.E.); martien.poelen@rivm.nl (M.C.M.P.); cecile.van.els@rivm.nl (C.A.C.M.v.E.); d.van.baarle@umcg.nl (D.v.B.); patricia.kaaijk@rivm.nl (P.K.); 2Center for Translational Immunology, University Medical Center Utrecht, 3584 CX Utrecht, The Netherlands; j.borghans@umcutrecht.nl; 3Department of Biomolecular Health Sciences, Faculty of Veterinary Medicine, Utrecht University, 3584 CX Utrecht, The Netherlands

**Keywords:** mumps infection, T-cell immunity, MMR vaccination

## Abstract

Waning of the mumps virus (MuV)-specific humoral response after vaccination has been suggested as a cause for recent mumps outbreaks in vaccinated young adults, although it cannot explain all cases. Moreover, CD8^+^ T cells may play an important role in the response against MuV; however, little is known about the characteristics and dynamics of the MuV-specific CD8^+^ T-cell response after MuV infection. Here, we had the opportunity to follow the CD8^+^ T-cell response to three recently identified HLA-A2*02:01-restricted MuV-specific epitopes from 1.5 to 36 months post-MuV infection in five previously vaccinated and three unvaccinated individuals. The infection-induced CD8^+^ T-cell response was dominated by T cells specific for the ALDQTDIRV and LLDSSTTRV epitopes, while the response to the GLMEGQIVSV epitope was subdominant. MuV-specific CD8^+^ T-cell frequencies in the blood declined between 1.5 and 9 months after infection. This decline was not explained by changes in the expression of inhibitory receptors or homing markers. Despite the ongoing changes in the frequencies and phenotype of MuV-specific CD8^+^ T cells, TCRβ analyses revealed a stable MuV-specific T-cell repertoire over time. These insights in the maintenance of the cellular response against mumps may provide hallmarks for optimizing vaccination strategies towards a long-term cellular memory response.

## 1. Introduction

Mumps is a viral infectious disease typically characterized by bilateral or unilateral swelling of the parotid glands. In some individuals, more severe complications, such as orchitis, deafness, meningitis, and encephalitis, occur [1]. Therefore, many countries vaccinate their population against mumps, usually as a combination vaccine together with measles and rubella vaccine components (MMR vaccine) [2,3]. This has led to a dramatic decrease in the incidence of mumps virus (MuV) infection [2]. However, in the last decades, mumps outbreaks have been reported in various countries, despite high vaccination coverage [4,5]. In the Netherlands, several mumps outbreaks occurred between 2009 and 2013, mainly among young adults, most of which did receive their two childhood MMR vaccinations [6].

Just as in natural infection, both humoral and cellular responses are induced after MuV vaccination [7,8]. Humoral responses have been investigated extensively, and one of the causes of vaccine failure in young adults is thought to be waning of antibody levels, although it may not explain all cases [3]. Another explanation for re-emergence of mumps could be the antigenic mismatch between the vaccine and outbreak strains [9,10]. The current Jeryl Lynn vaccine carries two viral isolates (JL-2 and JL-5) of the genotype A, whereas genotype G is the most recently circulating infectious mumps strain in the Netherlands [11,12]. These genotypes have a relatively large phylogenetic distance, with known antigenic differences [13,14]. CD8^+^ T cells are known to play an important role in the clearance of viruses and in disease outcome, and generally recognize the more conserved parts of a virus. Despite some mismatch epitopes between the vaccine and circulating MuV strains [15], many CD8^+^ epitopes are conserved, and may therefore play an important role in the prevention of vaccine failure, despite antigen mismatch. However, current insights into the MuV-specific cellular response are limited.

Studies focusing on the T-cell response against mumps have mostly investigated the response after MuV vaccination; however, it is generally observed that vaccination-acquired immunity is not as efficient as the long-term response induced after a natural infection [16,17,18]. This is also the case for the T-cell responses to MuV vaccination and MuV infection. Although MuV-specific T-cell proliferation and IFNγ production were detectable up to 21 years after vaccination, these responses were less pronounced than in naturally infected individuals [7,19,20]. It was also reported that MuV-specific CD8^+^ T cells after vaccination are less polyfunctional compared to those in naturally infected individuals [21]. Together, these studies suggest that current MuV vaccination evokes a less optimal cellular immune response compared to natural MuV infection. One way to prevent MuV vaccine failure could be to induce a sustainable T-cell response more comparable to the response against natural infection. To this end, more knowledge is needed about the clonal dynamics and characteristics of the MuV-specific T cells in time after natural MuV infection.

In this study, we had the unique opportunity to investigate in great detail the MuV-specific CD8^+^ T-cell response after natural MuV infection in five previously vaccinated and three unvaccinated individuals. Our group recently identified HLA-A*02:01-restricted MuV-specific T-cell epitopes from the recent outbreak strain (genotype G) [22]. Here, we ex vivo analyzed MuV-specific CD8^+^ T cells in mumps cases after infection using recently developed dextramers with which high frequencies of MuV-specific CD8^+^ T cells could be detected. We show that relatively large MuV-specific T-cell frequencies in the blood contracted significantly between 1.5 and 9 months after infection, to gradually further decline to low frequencies after up to 36 months. Although we observed changes in both PD-1 and CXCR4 expression between 1.5 and 9 months post-infection, the expression of these T-cell inhibitory and homing markers 1.5 months post-infection did not explain the decline in MuV-specific T-cell frequencies in the convalescence phase. The largest fraction of cells had a central memory phenotype (CD27+, CD45RO+) that remained stable over time, whereas a shift was found to memory precursor cells based on CD127 and KLRG-1 over time after infection. Despite the observed changes, a relatively stable T-cell receptor (TCR) repertoire in both vaccinated and unvaccinated individuals was found, indicating a sustainable T-cell response has been maintained in the long run.

## 2. Materials and Methods

### 2.1. Study Design

Samples were collected from mumps cases at 1.5 months, 9 months, 18 months, and 36 months after infection in two Dutch observational longitudinal studies, VAC263 [8] and IMMfact [21]. Written informed consent was obtained from all participants. All trial-related activities were conducted according to Good Clinical Practice, which includes the provisions of the Declaration of Helsinki. The studies were approved by the ethical committee METC Noord Holland and Review Board METC UMC Utrecht, respectively (clinical study numbers NL37852.094.11 and NL4679.094.13). Eight HLA-A2-positive mumps cases (21–53 years of age) were selected based on serotyping. Of these 8 subjects, 3 subjects were unvaccinated, while the other 5 had been vaccinated with two doses of the MMR vaccine during childhood. IgG titers were adopted from the study of Kaaijk et al. [8].

### 2.2. PBMC Isolation

Peripheral blood mononuclear cells (PBMCs) were obtained by Lymphoprep (Progen) density gradient centrifugation from heparinized blood, according to the manufacturer’s instructions. PBMCs were frozen in 90% fetal calf serum and 10% dimethyl sulfoxide at −135 °C until further use.

### 2.3. Analysis of MuV-Specific T Cells by Flow Cytometry

In the 8 HLA-A2-positive mumps cases, MuV-epitope-specific T-cell responses were analyzed by staining 4 million PBMCs in FACS buffer (2 mM EDTA, 0.5% BSA in PBS) using HLA class I dextramers for the epitopes ALDQTDIRV (ALD) of the M-protein (A*02:01/ ALD-FITC), GLMEGQIVSV (GLM) of the F-protein (A*02:01/ GLM-APC), and LLDSSTTRV (LLD) of the HN-protein (A*02:01/ LLD-PE) for 20 min at room temperature. Next, surface staining was performed in FACS buffer for 30 min at 4 °C, using the following monoclonal antibody (mAb) mix: CD27(M-T271)-PerCP-Cy5.5, CD3(SK7)-APC-R700, CCR7(150503)-BrilliantViolet711, CD45RO(UCHL1)-BrilliantUV395, CD4(SK3)-BrilliantUV737, CD183/CXCR3(1C6)-BrilliantViolet421 (all BD), CD8a(RPA-T8)-BV510 (BioLegend), CD127(A019D5)-BrilliantViolet650 (BioLegend), CD184/CXCR4(12G5)-BrilliantViolet786 (BD), Fixable Viability staining-780, and KLRG1(13F12F2)-PE-Cyanine7 (eBioscience). Acquisition was performed on an LSRFortessaX20 and data analysis was performed using FlowJo (Treestar). Populations of antigen-specific cells that were smaller than 30 events were excluded from further analysis. The gating strategy is shown in Appendix A.

### 2.4. Isolation of MuV-Specific T Cells for TCRβ Analysis

For cell sorting, cells were stained using dextramers and the following mAb mix: CD3(SK7)-APC-R700, CD4(SK3)-BriliantViolet711 (both BD), and CD8a(RPA-T8)-BrilliantViolet510 (Biolegend). CD3^+^CD4^-^CD8^+^dextramer^+^ cells were sorted on an FACSAria III directly into fetal calf serum (FCS) precoated tubes containing RNAlater (Ambion Inc. Applied Biosystems) and stored at −80 °C for subsequent TCRβ clonotype analysis. In addition, we stained with the following mAbs for further phenotypical analysis of the dextramer+ T cells: CD152/CTLA4(BNI3)-BrilliantViolet786, Fixable Viability staining-780 (both BD), PD1(EH12.2H7)-PerCP Cy5.5, Tim3(F38-2E2)-BrilantViolet421, (both BioLegend), and TIGIT(MBSA43)-PE Cyanine7 (eBioscience). Due to limited amounts of PBMCs, the phenotypic analyses of the MuV-specific T cells were combined with cell sorting for 3 individuals (i.e., subject 12 of the vaccinated individuals and subjects 02 and 10 of the unvaccinated individuals). This led to a less extensive phenotyping, including CD27(M-T271)-PerCP Cy5.5, CD3(SK7)-APC-R700, CCR7(150503)-BrilliantViolet711, CD45RO(UCHL1)-BrilliantViolet421, CD4(SK3)-BrilliantViolet786 (SK3), and Fixable Viability staining-780 (all BD), CD8a(RPA-T8)-BrilliantViolet510, CD127(A019D5)-BrilliantViolet650 (both BioLegend), and KLRG1(13F12F2)-PE-Cyanine7 (eBioscience). Again, populations of antigen-specific cells that were smaller than 30 events were excluded from further analysis.

### 2.5. Preparing TCRβ cDNA Libraries for Sequencing

T-cell receptor analysis was performed as described previously [23], with minor modifications. Briefly, mRNA was isolated with the RNA microkit (Qiagen) according to the manufacturer’s protocol. Isolated mRNA was used for cDNA synthesis with 5′RACE template switch technology to introduce universal primer binding sites, and unique molecular identifiers (UMI’s) were added at the 5′ end of the cDNA molecules using the SMARTScribe reverse transcriptase (TaKaRa). cDNA synthesis was followed by an AMPure XP bead-based clean-up (Beckman Coulter). Purified cDNA molecules were amplified in two subsequent PCR steps using the Q5^®^ High-Fidelity DNA Polymerase (New England BioLabs), with an AMPure XP bead-based clean-up step in between. PCR products were size-selected on gel and purified using the Nucleospin PCR clean-up kit (Machery-Nagel). The PCR products were sequenced via Illumina MiSeq paired-end 2 × 250 nt sequencing.

### 2.6. TCRβ Clonotype Analysis

The raw sequence data were processed using the 12 nt UMIs to correct for sequencing errors and unequal PCR amplification. RTCR [24] was used to identify both the UMI sequence and clonotype information from the reads. An additional filtering step was performed to exclude TCR sequences that were likely due to contamination in the sequencing protocol, and to minimize biases introduced by errors in the UMI sequence. In short, sequences were only accepted if their UMI was observed in at least 40 sequencing reads. Sequences with identical UMIs in multiple samples were removed if they did not occur in at least 1000 sequencing reads, and also if their absolute frequency was lower than 10% of the maximum frequency in the other sample. Within each sample, UMIs within a Hamming distance of 3 were clustered. More detailed information about the processing and filtering of the sequence reads is explained in Lanfermeijer et al. [25].

### 2.7. Alignment of MuV Peptides

Sequences for the alignment of the epitopes were obtained via uniprot.org. The corresponding Genbank accession numbers are: AF345290 for Jeryl Lynn 2, AF338106.1 for Jeryl Lynn 5, JX287390 for Genotype G5, JX287385.1 for Genotype G06, KY969483.1 for Genotype H, and MH892406 for Rubulavirus 2.

### 2.8. Statistical Analysis of Flow Cytometry Data

Differences between the groups (for example, vaccinated versus unvaccinated) were assessed using Mann–Whitney U-tests. Paired data (differences between timepoints or differences between epitopes) were compared using the Wilcoxon rank test (nonparametric).

Correlations were tested with Spearman’s rank correlation coefficient. For all analyses, *p*-values < 0.05 were considered statistically significant. Data were analyzed using GraphPad Prism 8.3 and SPSS statistics 22 for Windows (SPSS Inc., Chicago, IL, USA).

## 3. Results

### 3.1. Characteristics of Study Population

From a cohort of MuV-infected individuals, a total of eight HLA-A2-positive individuals were selected for this study. Five out of these eight cases received two doses of the MMR vaccine during childhood (at 14 months and 9 years of age), while the other three subjects were unvaccinated. In sera of all subjects, MuV-specific IgG antibodies were found (Table 1). The unvaccinated individuals had significantly lower IgG antibody concentrations 1.5 months after infection than the vaccinated individuals (*p* = 0.0357) (Appendix A), but this difference was neither present at 9 months post-infection, nor in the larger cohort [8]. There was no significant difference in severity of symptoms related to MuV infection between vaccinated and unvaccinated individuals (Table 1).

### 3.2. Decrease in MuV-Specific CD8^+^ T-Cell Frequencies over Time after Infection

Previously, our laboratory identified several MuV-specific HLA-A*02:01-restricted epitopes, including the matrix-protein-derived peptide ALDQTDIRV (ALD, residues 10–116), the fusion-protein-derived peptide GLMEGQIVSV (GLM, residues 253–262), and the hemagglutinin-derived peptide LLDSSTTRV (LLD, residues 505–513) (Appendix A). These three MuV-specific epitopes are immunogenic, as illustrated by relatively high frequencies of specific CD8^+^ T cells in MuV-infected individuals [22], and are conserved between several mumps strains, including the vaccine strains and the circulating (genotype G) outbreak strain (Appendix A). Dextramers loaded with these MuV peptides were used to detect MuV-specific CD8^+^ T cells after infection in previously vaccinated and unvaccinated mumps cases (Figure 1A). The dextramers allowed us to analyze the MuV-specific T-cell response at the epitope-specific level in depth at several timepoints after infection, starting at 1.5 months after MuV infection. Percentages of MuV-specific T cells in the blood decreased significantly over time after infection. Between timepoints 1.5 and 9 months after infection, the largest decrease in MuV-specific T-cell frequencies against all three epitopes was observed (*p* = 0.0078) (Figure 1B). MuV-specific T-cell frequencies remained low from 9 months on, but were still detectable up to 36 months after infection. No link between the decline in the MuV-specific T-cell response and waning of the antibody titers could be observed, whereas age also did not influence the frequencies of MuV-specific T cells (data not shown). No major differences in the magnitude or contraction of the MuV-specific T-cell frequencies were observed between previously vaccinated and unvaccinated individuals.

From peptide elution experiments with HLA-A*02:01-positive antigen-presenting cells, it is known that the abundance at which these peptides are presented by the HLA-A*02:01 molecule differs greatly between these three epitopes [26]. While the ALD epitope was found to be presented at the highest abundance, the GLM epitope was predicted to have the strongest binding to the HLA-A*02:01 molecule (Appendix A). We wondered whether this would influence the immunodominance of the CD8^+^ T-cell response induced by these three peptides. At 1.5 months after infection, T-cell frequencies against the LLD epitope (HN-protein) and the ALD epitope (M-protein) were significantly higher than those against the GLM epitope (F-protein) (Appendix A). By 9 months post-infection, these differences between T-cell frequencies against the different epitopes had disappeared. We also plotted the relative contribution of T-cell frequencies against the ALD, GLM, and LLD epitopes to the “total” MuV-specific T-cell response (sum of the T-cell frequencies induced by the three epitopes) (Figure 1C). In five out of eight mumps cases, the T-cell response against the ALD epitope was the most dominant response, while, in the other three cases, the response against LLD was dominant. The GLM peptide induced a subdominant response in all MuV-infected cases. We observed no significant difference in immunodominance between vaccinated and unvaccinated individuals and no obvious shifts in dominance over time after infection (Figure 1C). Thus, although the MuV-specific T-cell frequencies clearly contracted between 1.5 and 9 months after MuV infection, this did not impact the relative dominance of the specific T-cell responses within individuals.

### 3.3. Phenotype of MuV-Specific CD8^+^ T Cells Shifts from Short-Lived Effector to Memory Cells

We next investigated the maintenance of the phenotype of MuV-specific memory CD8^+^ T cells over time. To this end, we gated the memory T-cell subsets based on the expression of CD27 and CD45RO (gating strategy is shown in Appendix A). Every individual showed a consistent pattern in the distribution of the subsets for the three epitopes; however, the patterns differed between individuals (Appendix A). In general, the largest fraction of cells had a central memory phenotype (CD27^+^, CD45RO^+^), while, for the other memory subsets, large variations were observed between donors.

Focusing on the memory T-cell subsets based on the expression of CD127 (IL-7Ra) and killer cell lectin-like receptor G1 (KLRG-1), showed a different pattern (Figure 2A and Appendix A). The markers CD127 and KLRG-1 distinguish between early effector cells (EEC; CD127^−^, KLRG-1^−^), short-lived effector cells (SLEC; CD127^−^, KLRG-1^+^), double-positive effector cells (DPEC; CD127^+^, KLRG-1^+^), and memory precursor effector cells (MPEC; CD127^+^, KLRG-1^−^). Overall, between 1.5 and 9 months after infection, the fraction of EEC and SLEC within the MuV-specific T cells decreased (*p* = 0.0267 for EEC and *p* = 0.079 for SLEC), while the fraction of DPEC and MPEC increased (*p* = 0.0038 for DPEC and *p* = 0.0017 for MPEC) (Figure 2B). The fraction of MPEC remained rather low for all three peptides at all timepoints. For both gating strategies, we found no significant differences in the composition of the MuV-specific T-cell pool between vaccinated and unvaccinated, as well as no effect of age.

### 3.4. The Decline in MuV-Specific CD8^+^ T-Cell Frequencies Is Not Explained by the Expression of Inhibitory Markers

In chronic infections, the expression of inhibitory receptors on the cell surface of virus-specific CD8^+^ T cells is mostly associated with functional exhaustion, due to continuous antigenic stimulation [27,28]. The exact role of inhibitory receptors in acute infections remains unclear, but it has previously been suggested that these receptors regulate the primary response [29,30,31] and may limit immunopathology. Especially upregulation of PD-1 has been observed in response to activation of virus-specific T cells [32], but insight in the expression at later timepoints after infection are missing. Here, we measured the fraction of MuV-specific T cells expressing the inhibitory receptors PD-1, TIM3, or TIGIT, depicted above the graphs.

Despite the large variation in the percentage of PD-1-expressing MuV-specific CD8^+^ T cells between donors and between epitope specificities, the percentage of PD-1-expressing MuV-specific T cells was relatively high at 1.5 months after MuV infection (average of 50.78% ± 24.84%) (Appendix A, gating of a representative donor is shown in Appendix A) and significantly lower at 9 months post-infection (average of 43.10% ± 15.76). The percentage of both TIGIT^+^ and TIM3^+^ MuV-specific CD8^+^ T cells did not change significantly between these two timepoints (Appendix A). We observed no significant differences in the expression of these inhibitory receptors between vaccinated and unvaccinated individuals. In line with the study of Ahn et al. [33], the percentage of PD-1^+^ MuV-specific T cells was associated with the differentiation status of the MuV-specific T cells, as we found a significant negative correlation with the percentage of both CM (CD27^+^, CD45RO^+^) and MPEC (KLRG-1^−^, CD127^+^) at 1.5 and 9 months post-MuV-infection (Appendix A). Furthermore, the percentage of PD-1+ MuV-specific T cells showed a negative correlation with the TEMRA (CD27^−^, CD45RO^−^) phenotype, whereas no correlation with the EM (CD27^−^, CD45RO^+^) phenotype was observed.

To investigate whether there could be a role of these inhibitory receptors in restraining the response, we studied whether the decline in MuV-specific T cells between 1.5 and 9 months post-infection, indicated by the fold change in MuV-specific T-cell frequencies, correlated with the expression of inhibitory receptors. We hypothesized that, if inhibitory receptors lead to stronger contraction of the immune response, the higher expression levels of inhibitory receptors at 1.5 months post-infection should associate with a stronger decline in T-cell frequencies. In contrast, we observed a significant positive association between the expression of PD-1 at 1.5 months post-infection and the fold change in the height of the T-cell response (*r* = 0.7273, *p* = 0.0096), meaning that the MuV-specific T-cell responses with the lowest level of PD-1 expression 1.5 months after infection contracted the most (Appendix A). For TIGIT and TIM-3, no correlation between the expression of the receptors at 1.5 months post-infection and the decrease in MuV-specific frequencies was observed (data not shown). Although the expression of PD-1 decreased over time post-MuV-infection, the role of the expression of inhibitory markers after the acute phase of the immune response remains to be elucidated.

### 3.5. The Decline in MuV-Specific CD8^+^ T-Cell Frequencies Is Not Explained by Increased Expression of Bone Marrow Homing Markers

It has been suggested that virus-specific memory T cells, including those against MuV, are resting in the bone marrow for prolonged periods of time, even when their frequencies in the blood have already declined [34]. To study the migration of the MuV-specific T cells out of the blood towards other tissues, we measured the expression (based on mean fluorescent intensity) of the T-cell migration-associated markers CCR7, CXCR3, and CXCR4 on MuV-specific CD8^+^ T cells. CCR7 drives homing of T cells towards secondary lymphoid organs (SLO) [35], CXCR3 to the site of inflammation [36], and CXCR4 facilitates homing to the bone marrow [37]. The expression of both CCR7 and CXCR4 increased significantly between 1.5 and 9 months post-infection for T cells specific for all three epitopes (*p* = 0.0002 and *p* = 0.0137, respectively) (Figure 3A, left and middle panel), while the expression of CXCR3 remained stable between these two timepoints (Figure 3A, right panel).

Next, we investigated whether there was an association between the expression of tissue homing markers and the decline in T-cell frequencies in the blood between 1.5 and 9 months post-infection. Our hypothesis was that, if homing to the tissues would play a role in the decline in T-cell frequencies in the blood, high expression of the homing markers at 1.5 months post-infection would be associated with a stronger decline in MuV-specific T-cell frequencies for that person at 9 months post-infection. Indeed, CXCR3 expression at 1.5 months post-infection did show that the higher the expression of the homing marker, the stronger the decline in T-cell frequencies. In contrast, the lower the expression of CXCR4 at 1.5 months post-infection, the stronger the decline in MuV-specific T-cell frequencies observed (*r* = 0.7273, *p* = 0.0144) (Figure 3B). No association between CCR7 expression at 1.5 months post-infection and the decrease in MuV-specific CD8^+^ T-cell frequencies was observed. Thus, the high expression of CXCR3 and low levels of CXCR4 at 1.5 months post-infection associate with the decrease in T-cell frequencies, which implies migration to the inflammation site and bone marrow at 1.5 month. At 9 months, only CXCR4 expression is enhanced, which suggests migration to the bone marrow.

### 3.6. MuV-Specific TCRβ Repertoire Is Maintained in the Memory Phase

To investigate how the MuV-specific T-cell repertoire evolves over time after MuV infection, we were able to analyze the T-cell receptor (TCR) sequences of MuV-specific T cells in the blood samples over time. By sequencing the TCRβ chain of ALD-specific, GLM-specific, and LLD-specific T cells, we identified the variable (Vβ) and joining (Jβ) segment and the CDR3 region of the MuV-specific T-cell receptors (all identified TCR sequences are provided in Appendix A). At 1.5 months post-MuV-infection, we observed a dominant usage of the Vβ segment 15 (TRBV15) for the LLD epitope (87.7%, combined data of six donors) (Figure 4A, left panel). This large abundance is not only due to clonal expansions, as a relatively high percentage (38,5%) of the different TCR sequences specific for LLD also contained this segment (Figure 4A right panel). The data on the V segment usage of T cells recognizing the ALD and GLM epitopes were less conclusive, as these were based on fewer TCR sequences.

The T-cell repertoires against the three MuV-specific epitopes that we analyzed were relatively stable between 1.5 and 9 months after MuV infection, in that the same TCRβ sequences were found at both timepoints (Figure 4B,C, Appendix A). Vaccination status did not influence this, as this was observed in both vaccinated (Figure 4B) and unvaccinated individuals (Figure 4C). In summary, although overall MuV-specific T-cell frequencies in the blood clearly decreased between 1.5 and 9 months post-MuV-infection, we found no evidence that this resulted in loss of specific clones between these two timepoints.

## 4. Discussion

In this study, we investigated the frequency, phenotype, and TCR repertoire of the MuV-specific CD8^+^ T-cell response in the memory phase from 1.5 to 36 months after the onset of natural MuV infection in previously vaccinated and unvaccinated adults. We focused on the T-cell response against three recently identified HLA-A2-restricted epitopes of mumps virus [22]. We found a significant decline in MuV-specific T-cell frequencies against all three peptides in the blood between 1.5 and 9 months after infection, whereas the frequencies remained stable up to 36 months. Phenotypically, changes in PD-1 expression and in expression of memory and homing markers were observed, but none of these changes could be linked to the decrease in T-cell frequencies, except for CXCR3.

The CXCR3 expression was found to be associated with fold change of MuV-specific T-cell frequencies over time after infection. Despite the observed changes in T-cell frequencies and phenotype, the characteristics of the MuV-specific CD8^+^ T-cell response remained relatively stable at the TCRβ repertoire level.

We observed no obvious differences between childhood vaccinated and unvaccinated individuals in terms of T-cell frequencies, phenotype, and stability at the clonal level after MuV infection. Although T-cell responses have been observed up to 21 years after vaccination [19], the frequency and polyfunctionality of the CD8^+^ T-cell response against mumps has previously been described to be suboptimal after vaccination, as compared to the response after MuV infection [21]. Since we could not determine MuV-specific T-cell frequencies and characteristics prior to infection, we can only speculate about the presence of vaccine-induced memory T-cell immunity before infection in the vaccinated cases. Although the number of mumps cases analyzed in this study is small, the data suggest that the course of the CD8^+^ T-cell response after MuV infection is similar in unvaccinated and childhood vaccinated adults. In addition, there was also no age-related correlation found in the data.

The three MuV-specific epitopes used in this study are derived from different MuV-proteins. Nevertheless, the dynamics and characteristics of the T-cell responses against these three epitopes were comparable. There was only a difference in immunodominance, as we consistently found higher LLD-specific compared to GLM-specific CD8^+^ T-cell frequencies, both 1.5 months and 9 months post-MuV infection. T-cell dominance did not seem to be influenced by the binding strength of the peptides to the HLA-A2:01 molecule (which was highest for the GLM peptide) or the abundance of the peptide (which was highest for the ALD peptide), as the response against the LLD epitope was the most dominant response (Appendix A).

The decrease in MuV-specific T-cell frequencies that we observed between 1.5 and 9 months after MuV infection was similar for vaccinated and unvaccinated, and occurred at a relatively later stage of the contraction phase. The peak of expansion and subsequent start of the contraction most likely took place at an earlier timepoint after infection [38,39]. It is generally assumed that T-cell frequencies after acute infection stabilize within 30 days post-infection [40]. However, the later phase of the T-cell response after acute infection has not frequently been explored and the exact length of the contraction phase remains to be elucidated. Therefore, it remains unknown whether the observed contraction is part of the end of the classical contraction phase, or whether the decline after 1.5 months post-infection is specific for MuV infection and delayed compared to the contraction of the T-cell response after other acute infections. Studies comparing the expansion and contraction phase of primary and secondary responses against acute infections actually showed that the secondary T-cell response consisted of a prolonged contraction phase compared to the primary response [40,41]. A prolonged contraction phase may, thus, even result from involvement of the memory response. Based on the timepoints in this study this could, unfortunately, not be investigated. We also found no link between the contraction of the CD8^+^ T-cell response and waning of the antibodies after infection.

Although we did observe a decrease in the expression of the inhibitory marker PD-1 and an increase in the expression of the homing markers CXCR4 and CCR7 on MuV-specific CD8^+^ T cells over time, it is difficult to interpret the exact role of these markers in the memory phase after an acute infection. For PD-1, the observed decrease in expression could be a late effect of its early upregulation, which is often observed upon activation of T cells at the start of the response [32]. In case of the homing markers, we were surprised to find an upregulation of CCR7 between 1.5 and 9 months post-infection. We expected this marker to play a more dominant role during the acute phase of infection, as CCR7 is related to homing to secondary lymphoid organs, which most likely plays a role during T-cell priming [42,43].

We were interested in the expression of CXCR4, as previous studies have suggested that the bone marrow contains large fractions of memory T cells, including MuV-specific T cells [34,44,45,46]. These antigen-specific memory T cells in the bone marrow were shown to be resting and to be maintained for long periods of time, probably for decades, even when they were no longer detectable in the blood [34]. Surprisingly, we found that higher expression of CXCR4 at 1.5 months post-infection was associated with a lower decrease in MuV-specific T-cell frequencies in the blood. We cannot exclude the possibility that, in individuals with MuV-specific T cells with a relatively high level of CXCR4 expression, MuV-specific T cells were already migrating to the bone marrow before our first timepoint. Alternatively, homing of MuV-specific T cells to the bone marrow may have occurred between the two timepoints at 1.5 and 9 months post-infection, which would also explain the increased levels of CXCR4 expression at 9 months post-infection. As the time interval between the two timepoints is relatively large, more samples between 1.5 and 9 months would be needed to follow the dynamics of the expression of CXCR4 in relation to the decrease in MuV-specific frequencies.

Although no significant change in CXCR3 expression was observed between 1.5 and 9 months post-infection, we did find an association between the level of CXCR3 expression at 1.5 months post-infection and the fold decline in MuV-specific T-cell frequencies. CXCR3 expression is known to be associated with migration to the site of inflammation [47]. We may speculate that the T cells at 1.5 months still need to be present at the site of inflammation, whereas homing to the bone marrow probably takes place at a later timepoint.

The overall decrease in MuV-specific T-cell frequencies between 1.5 and 9 months fits well with the observed decrease in the EEC and SLEC fraction of the MuV-specific T cells over time, as these two populations play a role early after infection and, subsequently, contract [48]. However, it can be noted that the population of MuV-specific EECs and SLECs remains rather high at 9 months post-MuV infection. The relatively high frequency of short-lived and early effector cells seems a bit counterintuitive 9 months post-infection. The presence of high percentages of KLRG-1^+^, CD127^−^ (SLEC) in the memory phase has been described in chronic infection, but not for acute infections [48]. In mice, Renkema et al. showed that the KLRG-1^+^, CD127^−^ population did not contract after infection, and are actually long-lived effector cells (LLECs). This population was further characterized by lack of expression of CD62L, CD27, and CCR7 [49]. Analysis of KLRG-1^+^ MuV-specific T cells at 9 months indeed showed that part of the response lacks CD27 or CCR7 (data not shown). MPECs, on the other hand, have an increased propensity to persist in the memory phase. Indeed, MuV-specific T cells showed an increase in the fraction of MPECs over time, suggesting that these cells contribute to the MuV-specific long-term memory response. Therefore, the MuV-specific T cells at 9 months post-infection may be considered to have a more prominent memory and long-lived effector phenotype, rather than short-lived effector cells.

Despite the strong decrease in MuV-specific T-cell frequencies in the memory phase of the response to MuV infection, the MuV-specific TCR repertoire remained relatively stable between 1.5 and 9 months post-MuV-infection. Longitudinal data on the diversity and stability of TCR repertoire usage of virus-specific T cells after acute infection are limited. Vaccination with live-attenuated yellow fever virus (YFV) is often used as a model for acute infection, as it induces a strong T-cell response, which can be detected in the blood even decades after infection [50,51]. In a study focusing on the antigen-specific T-cell repertoire after YFV vaccination, it was shown in one donor that the yellow-fever-virus-specific T-cell receptor sequences observed 45 days after primary vaccination were comparable to those observed after secondary vaccination 18 months later, suggesting that practically all clones were maintained in the memory pool and responded during secondary vaccination [52]. Similar results have been observed after chronic infection with Epstein Barr virus (EBV). It was shown that a stable EBV-specific TCR repertoire was established after infection, as reflected by similar Vβ segment usage and recurrence of the same dominant T-cell clonotypes at different timepoints after the acute phase of the infection [53,54,55]. As EBV-specific T-cell frequencies remain relatively high during life, for EBV, it could be shown that the responding TCR repertoire remained stable for at least 3 years after infection [50,51,52]. Our data are in line with these studies and suggest that, after acute infection, the MuV-specific TCR repertoire is also stable for at least a few months after infection and during contraction.

As we observed a stable T-cell response against MuV, which was still detectable up to 36 months after infection, the question remains to what extent the induced MuV-specific CD8^+^ T cells after infection will contribute to protection against MuV infection in the long term. Due to the low number of MuV-specific T cells, we were unable to measure functional responses, which will be of great value for further studies. Another question that remains unanswered based on this data is why the vaccinated individuals in this cohort were infected and developed mumps disease. One possibility would be that the T-cell response induced by the MuV vaccine is not as stable as the response induced by MuV infection. Our earlier findings suggest that the relatively reduced magnitude and polyfunctionality of the MuV-vaccine-induced CD8^+^ T-cell response could play a role [21]. Insight into the pre-existing MuV-specific T cell response in vaccinated individuals would have been very informative. Unfortunately, in this study, we cannot assess the pre-existence of childhood MuV-vaccination-induced memory T cells, as pre-infection samples were not available.

Taken together, our data on the characterization of the T-cell responses of MuV-infected individuals suggest the development of a sustainable T-cell memory population. However, most of our adult mumps cases were childhood vaccinated. This raises the question of whether the vaccine-induced CD8^+^ T-cell response was still present in these individuals, as we found no clear differences in the response between vaccinated and unvaccinated individuals after MuV infection. Future studies should highlight how we could enhance the effectivity and longevity of the CD8^+^ T-cell response after MuV vaccination. The results of this study add to the body of knowledge on effectivity and longevity of CD8^+^ T-cell responses induced by natural infection, and may be helpful in optimizing vaccination strategies aimed at obtaining long-term cellular memory.

## Figures and Tables

**Figure 1 vaccines-09-01431-f001:**
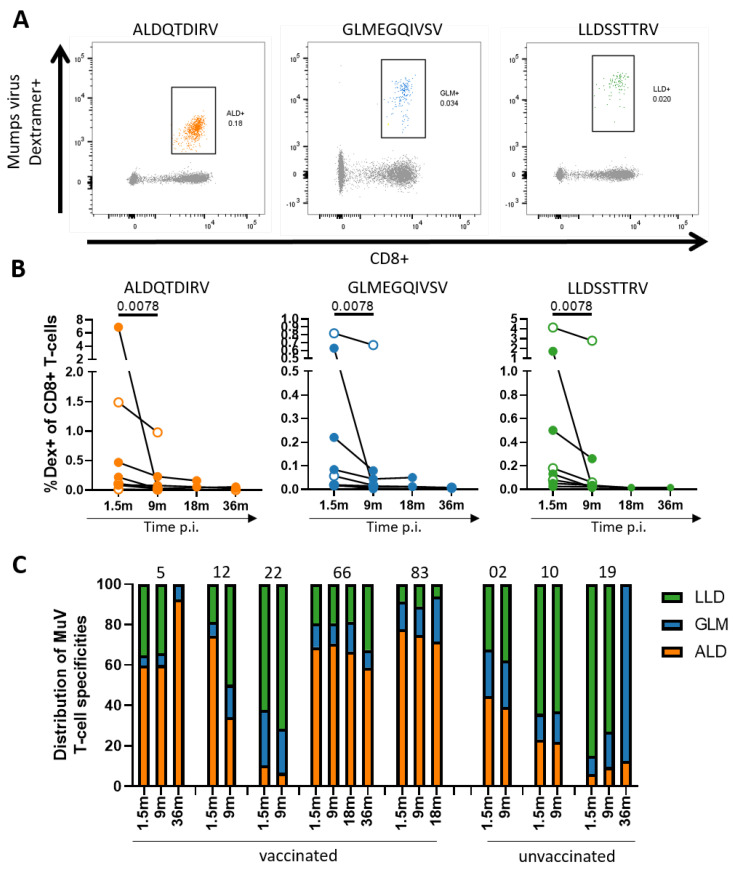
Frequencies of the MuV-specific CD8^+^ T cells wane after infection. (**A**) Representative dextramer staining and quantification of the MuV-specific CD8^+^ T cells against ALD (**left**), GLM (**middle**), and LLD (**right**). (**B**) Percentages of the MuV-specific CD8^+^ T cells against ALD (**left**), GLM (**middle**), and LLD (**right**) in HLA-A2-positive individuals upon MuV infection. Solid circles indicate vaccinated individuals, whereas open circles indicate unvaccinated individuals. p.i., post-infection. The percentage of dextramer^+^ CD8^+^ T cells of donor 5 have been published before in the study of de Wit et al. 2020 [22]. (**C**) The relative contribution of the three MuV epitopes to the “total” (=sum of the frequencies of the three) MuV-specific CD8^+^ T-cell response over time. Donor numbers are depicted above the graphs. Wilcoxon rank test was used to compare T-cell responses of individuals over time.

**Figure 2 vaccines-09-01431-f002:**
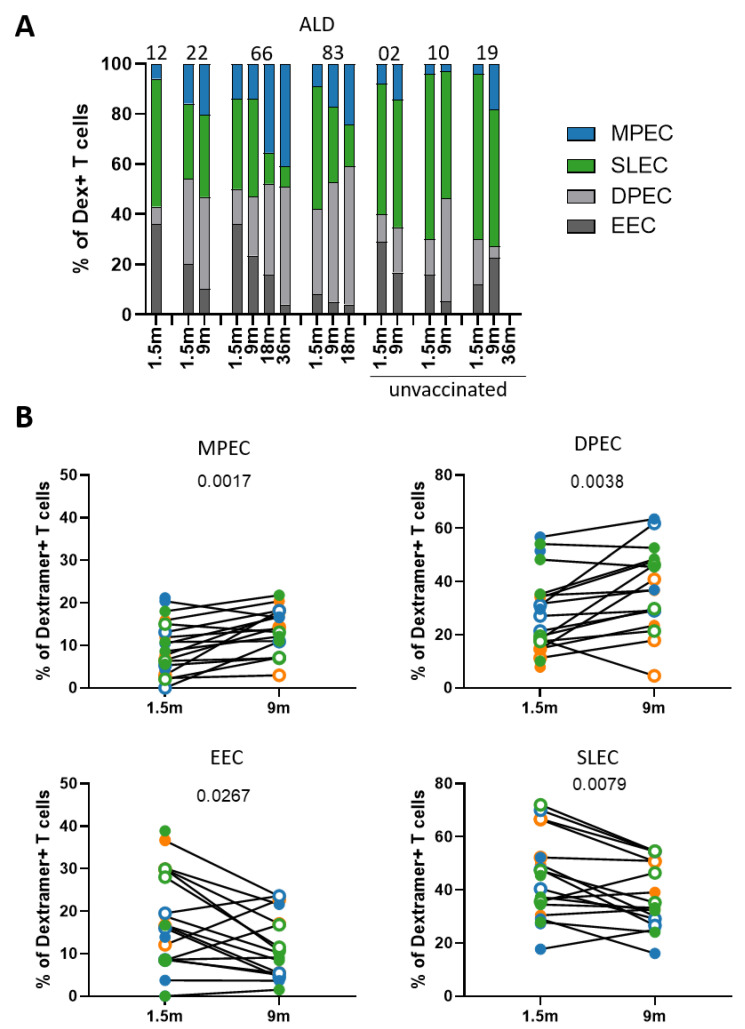
MuV-specific CD8^+^ T cells differentiate from effector cells towards memory cells over time after infection. (**A**) Bar graph showing the subset distribution based on CD127 and KLRG-1 expression of the MuV-specific CD8^+^ T cells against the ALD epitope. Donor numbers are depicted above the graphs. (**B**) Fraction of the memory subsets based on CD127 and KLRG-1 expression of the MuV-specific CD8^+^ T cells at 1.5 months and 9 months after MuV infection. The memory precursor cells (MPEC; CD127^+^, KLRG-1^−^), short-lived effector cells (SLEC; CD127^−^, KLRG-1^+^), double-positive cells (DPEC; CD127^+^, KLRG-1^+^), and the early effector cells (EEC; CD127^−^, KLRG-1^−^). CD8^+^ T cells specific for the ALD epitope are depicted in orange, for the GLM epitope in blue, and for the LLD epitope in green. Solid circles indicate vaccinated individuals, whereas open circles indicate unvaccinated individuals. Differences between timepoints were tested by Wilcoxon rank test.

**Figure 3 vaccines-09-01431-f003:**
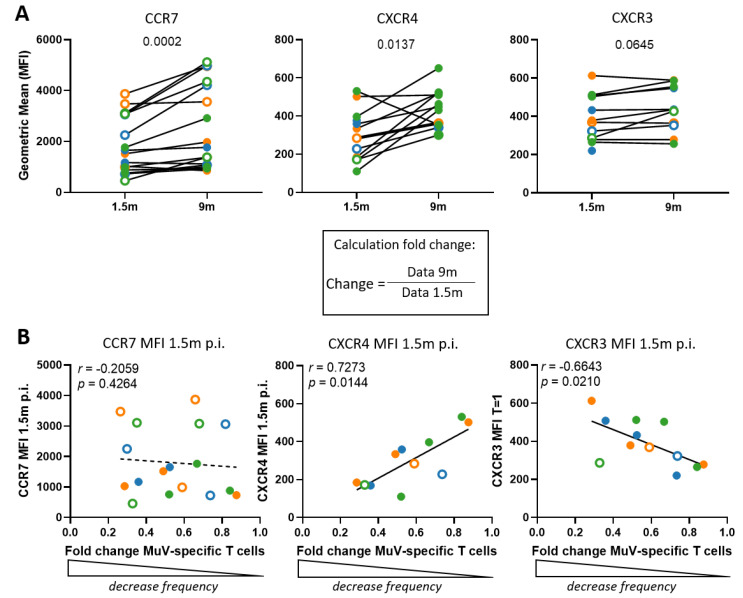
MuV-specific T cells tend to home to the bone marrow. (**A**) The expression of chemokine receptors CCR7 (**left**), CXCR4 (**middle**), and CXCR3 (**right**) based on their MFI value (geometric mean fluorescence intensity) of the MuV-specific T cells at 1.5 months and 9 months after MuV infection. (**B**) Association between the expression of CCR7 (**left**) or CXCR4 (**right**) (geometric mean of MFI) at timepoint 1.5 months post-infection and fold change of dextramer frequencies at timepoints 1.5 months and 9 months after infection. Fold changes were calculated by dividing the expression or frequencies found at 9 months after MuV infection by the expression or frequencies 1.5 months after MuV infection, the calculated fold changes were all below 1, indicating a decrease. Expression of chemokine receptors on CD8^+^ T cells specific for the ALD epitope are depicted in orange, for the GLM epitope in blue, and for the LLD epitope in green. Solid circles indicate vaccinated individuals, whereas open circles indicate unvaccinated individuals. Differences between timepoints were tested by Wilcoxon rank test. Associations were tested by Spearman’s correlation.

**Figure 4 vaccines-09-01431-f004:**
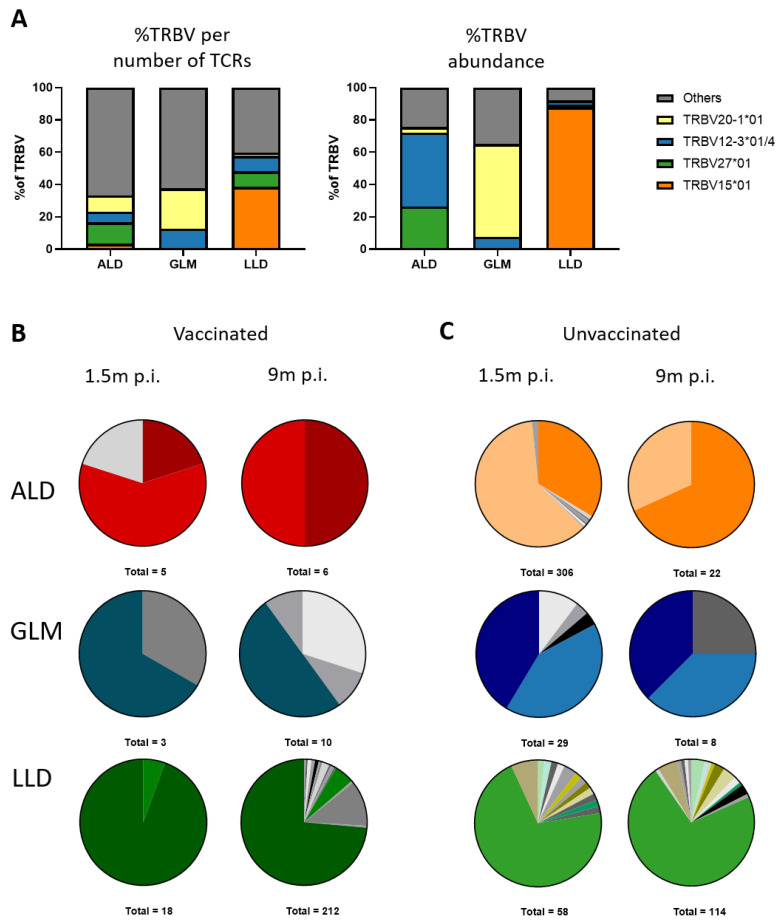
MuV-specific TCRβ repertoire is maintained in the memory phase. (**A**) Contribution of several Vβ families in the MuV-specific CD8^+^ T-cell repertoire based on number of sequences (**left** panel) and based on abundance (**right**) at 1.5 months of all donors (Appendix A). (**B**,**C**) Characterization of the T-cell repertoire of MuV-specific CD8^+^ T cells against ALD, GLM, and LLD, detected by PCR of vaccinated (**B**) and unvaccinated (**C**) individuals. Each pie chart depicts the repertoire of a representative donor at a certain timepoint (1.5 months and 9 months after infection). Colors represent shared CDR3 sequences between timepoints and donors. Grey scales depict unique CDR3 sequences. Total number below a pie indicates the number of clones detected. p.i., post-infection.

**Table 1 vaccines-09-01431-t001:** Study population.

Donor	Sex	Age (yrs)	Time Points	Vaccination Status	IgG Concentration, 1.5 Months after Infection (RU/mL)	IgG Concentration, 7–10 Months after Infection (RU/mL)	Clinical Symptoms
263-05	M	21	1.5 m, 9 m, and 36 m	Vaccinated	4436	3010	Parotitis, swollen neck glands, fever, cold, cough
263-12	M	25	1.5 m and 9 m	Vaccinated	7265	5928	Parotitis, swollen neck glands, fever
263-22	M	26	1.5 m and 9 m	Vaccinated	21,683	7108	Orchitis
274-66	F	30	1.5 m, 9 m, 18 m, and 36 m	Vaccinated	34,843	10,579	Parotitis, swollen neck glands
274-83	F	20	1.5 m, 9 m, and 18 m	Vaccinated	12,785	23,669	Parotitis, fever, permanent unilateral deafness
263-02	F	26	1.5 m and 9 m	Unvaccinated	358	1856	Parotitis, swollen neck glands, abdominal pain, cold, otitis
263-10	M	40	1.5 m and 9 m	Unvaccinated	2396	663	Orchitis, parotitis, swollen neck glands, fever, sore throat
263-19	F	53	1.5 m and 9 m	Unvaccinated	449	3704	Swollen neck glands, fever, cough, vertigo, temporary deafness

## Data Availability

The data presented in this study are available on request from the corresponding author, with consideration of the participants’ privacy rights.

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
