# Peer review of "Longitudinal Characterization of the Mumps-Specific HLA-A2 Restricted T-Cell Response after Mumps Virus Infection"

_vaccines, 2021, doi:10.3390/vaccines9121431_

Round 1

Reviewer 1 Report

In this paper, Josien Lanfermeijer and colleagues describe different features of HLA-A2*02:01-restricted Mumps virus (MuV)-specific T cells in a small longitudinal cohort of mumps-infected individuals. The relevance of understanding natural responses to MuV is clearly stated and the paper is well writen and presents the results in a logical sequence. Here are the main points of concern, according to this reviewer:
1)    Despite the fact that they have vaccinated and unvaccinated individuals in this cohort, meaningful comparison is precluded from the small sample size. 
2)    Although the complete cohort (including the individuals restricted by HLA-A2-02:01) is quite unique, another important limitation of this study is the absence of samples pre-infection, to be able to determine the presence, frequency and characteristics of MuV-specific T cells in vaccinated individuals.
3)    I have some concerns regarding the description and interpretation of the phenotype of tetramer+ MuV-specific T cells.
a.    Although the flow cytometry-based analysis was performed simultaneously, the authors present the data about memory/naive subsets in 2 ways: based on CD27 and CD45RO, (but leaving CCR7 out) or based on KLRG1 and CD127 with contradicting results. For example, while one analysis shows predominance of central memory, the other shows high abundance of EECs and SLECs. Also, although the authors show a decrease in EECs and SLECs at 9 months p.i., the proportion of these populations is still remarkable high within the total tetramer+ cells. This defies the definition of these subsets, which should be short lived and have an effector phenotype. I would suggest the authors combine CD27, CD45RO, CCR7, KLRG1 and CD127 into one analysis and check whether what they see fits with has been described for other viral infections. 

b.    All figures related to the phenotype of tetramer+ T cells fail to show representative dot plots of the data. Only bargraphs are shown. In order to understand the authors’ analysis, it would be very useful to see the gating strategies used. 

c.    Reference to Supplementary Fig 6A is missing in the text. It is quite counterintuitive that the expression of inhibitory receptors is so high after 9 months in a resolving acute infection. Also, the relationship between levels of PD-1 and contraction of the response is contrary to expected. Depiction of representative dot plots would again help interpretation. Levels of inhibitory receptors have been linked to differentiation status of the cells. Could the authors explain the variability in expression levels of these receptors to the composition of memory subsets for each patient?

d.    Analysis of chemokine receptor expression was based on MFI and not on percentage. However, expression of CCR7, CXCR3 and CXCR4 is usually dicotomic, with clear positive and negative populations. Also, it would be very interesting to look at coexpression patterns of these receptors and their correlation to naive and memory subsets. 

Reviewer 2 Report

Lanfemeijer and colleagues investigate the quantity and quality of mumps virus-specific CD8+ T cells derived from a small set of MMR-vaccinated and vaccine-naïve individuals previously infected with mumps virus. Using high dimensional flow cytometry, peptide-dextramers and TCR sequencing antigen-specific CD8+ T cells are characterized over time, from 1,5 months up to 36 months. Especially, the longitudinal follow-up is of significant interest to the field of vaccinology and T cell biology. The manuscript is very well written and the data is presented and described in an accurate manner. Shortcomings are excellently addressed in the discussion. However, the manuscript needs revisions in the results section to warrant the claims made by Lanfemeijer and colleagues.

Major concerns:

General: The authors should consider implementing several supplementary figures into the main figures. For example, in section 3.3 a significant conclusion on T cell subset contraction is based on supplementary figure 5, which would better fit in the main figure 2. Please reconsider the supplementary figures based on the text and its conclusions.

Gating strategies in supplementary data will significantly aid in understanding the robustness of the data presented in bar graphs.

In all experiments reporting on MFI, the expression of the non-specific CD8+ T cells in the same patient sample should be incorporated to ensure the biological effect (eg. Increase/decrease of marker MFI) is specific to the cell type and not a result of patient-specific expression levels. Especially in Figure 3B the MFI of antigen-specific CD8+ T cells is used as an independent value to correlate with the frequency decrease, which is appreciated, but can be alternatively explained by interpatient variation.

All relevant variables used for correlation analysis should be correlated with the patient age.

Methods: The major readout is flow cytometry using a BD Fortessa, which is subject to measurement changes/sensitivities over time, leading to batch effects. As such, to assess marker MFI over time (especially over like +12 months), adequately calibration needs to be assured. This includes laser/detector base line calibration, daily CST bead calibration and standardization of staining protocols/reagents.

Figure 1B. Reuse of published data should be clearly defined in the figure legend and the difference between novel and published data made explicit in the results section. Also, the added value of new data should be mentioned.

Supplementary Figure 3 and text (224-226); this conclusion is not warranted by the data, differences in p-values between 1.5m and 9m groups are a matter of standard deviation and not difference in mean.

Minor concerns:

Supplementary Figure 3 misses peptide definition in the color coding.

A link between antibody titers and cellular immunity would greatly enhance the understanding of the total decline in the immune response; is contraction of cellular immunity tied to antibody waning? I understand from Supplementary Figure 1 that adequate correlation analysis will be difficult because of the low number of patients included (and spread between groups), but addressing the question in the results or discussion section would benefit the manuscript.

Reviewer 3 Report

In this study, the authors assessed the frequency, phenotype, and TCRβ repertoire of the MuV-specific CD8+ T-cell (CTL) response against three HLA-A2*02:01-restricted MuV epitopes during the memory phase (1.5 to 36 months) after MuV-infection in eight HLA-A2-positive subjects—three unvaccinated and five childhood-vaccinated adults. The authors report that the CTL frequencies and phenotype varied in both the groups of study subjects and that the TCRβ repertoire level remained relatively stable in both these groups. Interestingly, between the vaccinated and unvaccinated study subjects, there were no significant differences: in the magnitude or contraction of the MuV-specific T-cell frequencies; in the composition of the MuV-specific T-cell pool; and in the stability of the TCRβ repertoire. The authors conclude that the course of the CTL response to MuV infection was comparable between the two groups of study subjects.

As the authors point out, several interesting and important questions remain unanswered, including whether the vaccine-induced CTL response was still active (and, if yes, how robust was it) in the study subjects vaccinated as children. While the study is limited to HLA-A*02:01-restricted MuV epitopes and the findings do not provide clear insights into the molecular mechanisms driving the variations in T-cell frequencies and phenotype, the finding of the presence of a stable MuV-specific TCR repertoire is noteworthy.

Any revision of the manuscript should address the major and minor concerns listed below so as to meet the Journal’s requirement for publication.

MAJOR POINTS:

1) The title “Longitudinal characterization of the mumps-specific CD8+ T-cell response after mumps virus infection” is generic and ambiguous. This study assessed specifically the CD8+ T-cell response against the HLA-A*02:01-restricted MuV-specific epitopes. Therefore, consider a specific title that accurately reflects the major finding(s) reported in the manuscript (for instance, by denoting HLA-A2 positive study subjects in the title).

2) The small sample size is certainly a major concern. The authors should consider specifying the number of study subjects in the abstract. For instance: “Here we had the opportunity to follow the CD8+ T-cell response to 3 recently identified HLA-A2*02:01-restricted MuV-specific epitopes from 1.5 till 36 months post MuV-infection in five previously vaccinated and three unvaccinated individuals.”

MINOR POINTS:

1) Table 1.

Comment: Check text alignment in the “Clinical Symptoms” column between different donors, especially in the case of vaccinated Donors 263-12 and 263-22 and in the case of the unvaccinated Donors.

2) Lines 211 and 317:

Comment: Consider “indicate” instead of “indicated”

3) Line 267: “… over time after infecTable 127.”

Comment: Please check for typographical errors here and elsewhere in the manuscript including the Supplementary Figure legends.

4) Line 271: “Donor numbers are.”

Comment: Incomplete sentence. The remaining part of the sentence seems to be in line 265.

5) Lines 287-289: “For TIGIT and TIM-3 no correlation between the expression of the receptors at 1.5 months post infection and the decrease in MuV-specific (data not shown).”

Comment: Please check for sentence fragments and run-on sentences here and elsewhere in the manuscript. Please rephrase for clarity.

6) Line 311: “… at timepoint 1.5 post infection …”

Comment: 1.5 months?

7) Line 325: “… did show the higher the expression …”

Comment: did show that?

8) Lines 502-509:

Comment: The institutional review board statement section should contain an applicable author statement. Instead, it contains only the journal’s instructions to the authors.

9) Lines 510-517:

Comment: The informed consent statement section should contain an applicable author statement. Instead, it contains only the journal’s instructions to the authors.

10) Lines 518-522:

Comment: The data availability statement section should contain an applicable author statement. Instead, it contains only the journal’s instructions to the authors.

Round 2

Reviewer 1 Report

The authors have gone through my comments and those of 2 additional reviewers and have modified the manuscript to incorporate some additional information. Of note, gating strategies and representative plots are a useful addition to the paper as they help assess the soundness of the data.

Major concern:

I still have concerns about the conclusions based on KLRG1 and CD127 expression. The discrepancy between the concept of SLECs (short-lived effector cells) and their frequencies at 9 months pi (when the response seems to be fully contracted) has not been fully explored.

At the same time, in the analysis based on CD45RO and CD27 expression, the authors find a prominent proportion of virus-specific CD8 T cells with a CM phenotype and an increase in CCR7 MFI at 9 months pi. These last results are in contradiction the idea that the cells defined here as SLECs at 9 months pi are in fact short-lived effectors. Both the timing and the phenotype would suggest otherwise.  

The authors state that the size of the tetramer population is too small for further dissection of virus-specific T cells. Is it perhaps possible to combine KLRG1 vs CCR7 and KLRG1 vs CD27 for at least the most abundant virus-specific populations? 

Could it be possible that these KLRG1+ CD127+ and KLRG1+ CD127- cells are actually long-lived instead of short-lived effectors? This concept has already been introduced for both mice (https://www.jimmunol.org/content/205/4/1059) and human (https://onlinelibrary.wiley.com/doi/10.1002/eji.201847897). 

Minor comment:

In Suppl. Fig. 8B, the plot of CD8 vs PD-1 at 1.5 and 9 months for the total CD8+ T cells seems to be duplicated. 

Reviewer 2 Report

The authors have addressed my concerns to my satisfaction and endorse accepting the manuscript.

Author Response

Thank you for the effort of reviewing our manuscript